# Highly Compressible Elastic Aerogel Spring-Based Piezoionic Self-Powering Pressure Sensor for Multifunctional Wearable Electronics

**DOI:** 10.3390/nano12152574

**Published:** 2022-07-27

**Authors:** Ning Wei, Yan Li, Chunqin Zhu, Yuxi Tang

**Affiliations:** 1School of Electronic Information and Electrical Engineering, Hefei Normal University, Hefei 230601, China; liyan010830@163.com (Y.L.); zcq4112020@163.com (C.Z.); sakitama077@163.com (Y.T.); 2AnHui Province Key Laboratory of Simulation and Design for Electronic Information System, Hefei Normal University, Hefei 230601, China

**Keywords:** self-powering, electronics, wearable, sensor

## Abstract

To meet the rapid development of wearable flexible electronics, the multifunctional integrations into singe device are in extreme demand. In this paper, we developed novel self-powering multifunctional pressure sensors and supercapacitor-integrated device based on highly elastic silver nanowires@reduced graphene aerogel, being conductive to reduce integration difficulties and device size. Serving as an energy device, it behaves with a prominent specific capacitance of 146.6 F g^−1^, and excellent rate capability even at 500 mV s^−1^. The fabricated sensor demonstrates an excellent sensitivity of 2.54 kPa^−1^ and superior pressure-sensing stability up to 1000 compressive cycles. Piezoionization effect is suggested to reveal the sensing mechanism. Our research provides a new research direction in designing the integration of self-driving wearable electronics.

## 1. Introduction

With the rapid development of electronic information devices, intelligent electronics have been applied in wearable sensors, energy harvesting devices, nanogenerators, human–machine interfaces, soft robotics, and healthcare monitoring sensors [1,2,3]. Recently, integrated multifunctional devices have extensively attracted great interest from researchers [4,5,6]. Thus, the seamless integration of multifunction into one device without increasing the size is highly desired.

Graphene composite aerogel-based pressure sensors and supercapacitors have been widely explored. Ge et al. have reported rGO/polyaniline wrapped sponge-based pressure sensor with tunable sensitivity (0.042 to 0.152 kPa^−1^) [7]. Chen et al. have prepared electron-induced perpendicular graphene electrodes for capacitive sensor with high sensitivities of 0.13 kPa^−1^ [8]. Dong et al. have synthesized pressure sensor based on a graphene/silver nanowires that exhibits a high sensitivity of 0.016 kPa^−1^ at pressure of 0–40 kPa [9]. Wei et al. have assembled resilient bismuthine-anchored graphene-based symmetric supercapacitor that delivers specific capacitance of 128.1 F g^−1^ at 0.5 A g^−1^ [10]. Yu et al. have reported functionalized graphene aerogel-based supercapacitor with the maximum specific capacitance of 108.9 F g^−1^ at a scan rate of 5 mV s^−1^ [11].

Recently, Zhu et al. adopted polyurethane foam to fabricate a dual-model sensor for both pressure and temperature [12]. Zhang et al. demonstrate a strain sensor based on CNTs@PDMS film as well as a flexible supercapacitor [13]. Zheng et al. have reported a highly integrated energy storage and pressure sensor system based on MXene-derived nanorods [14]. Zhao’s group showed a self-powering integrated system consisting of a flexible supercapacitor and additional pressure sensor [5]. This research has achieve the multifunction of electronic devices, while the energy storage and pressure sensing functions have been realized independently via different device structures. The implementation of various device structures increases the integration difficulty and device size. Hence, the integration of two functions into one device structure based on a single active material also remains challenging. Additionally, there are less reports on such single structure-based multifunctional devices.

To overcome these challenges, we design and fabricate a highly elastic silver nanowires@graphene aerogel spring-based device that can realize dual-mode pressure sensing and energy storage functions. Based on the self-discharging feather of a flexible all-solid-state symmetrical supercapacitor, a highly sensitive self-powering pressure sensor is realized simultaneously. Serving as an energy device, it behaves with prominent specific capacitance of 146.6 F g^−1^, together with excellent rate capability even at 500 mV s^−1^. The fabricated sensor shows excellent sensitivity of 2.54 kPa^−1^ in the pressure range of 0–3.6 kPa, and high strain sensing stability within 1000 compressing–releasing cycles.

## 2. Experiments

### 2.1. Synthesis of Silver Nanowires@Reduced Graphene Aerogel

The preparation process of the silver nanowires@reduced graphene aerogel (AgNWs@rGA) sample was achieved as follows. An aqueous graphene oxide (5 mg mL^−1^) solution was prepared by dissolving 200 mg GO in 40 mL deionized water, then were added 400 µL reducing agent ammonia and 4 mL of silver nanowire solution (5 mg mL^−1^), which is composed of silver nanowires with a diameter of 25–30 nm and length of 18–23 µm, under magnetic stirring for 30 min. A total of 4 mL mixed solutions was transferred into a 25 mL Telfon-lined autoclave at 120 ℃ for 6 h. The resulting hydrogel had an average height of 5.2 mm and diameter of about 14.5 mm. The obtained hydrogel was freeze-dried into a composite aerogel under vacuum of about 5 Pa and temperature of about −50 ℃ for 24 h. The aerogel exhibited with an average height of 4.6 mm and diameter of about 13.8 mm, denoted as AgNWs@rGA.

### 2.2. Preparation of Flexible Symmetrical Supercapacitor and Sensitive Piezoionic Self-Powering Sensor

The symmetrical supercapacitor was fabricated as follows. The all-solid-state supercapacitor was assembled by placing two slices of aerogel with equal mass of about 1.8 mg, which are cut off from the monoblock of AgNWs@rGA with electric resistance of ≈22.5 Ω. PVA-H_2_SO_4_ was used as gel electrolyte, which was prepared as follows: A total of 6 g of PVA is dissolved in 60 mL of water at 90 °C under stirring until solution became transparent. Then, dissolved PVA was added with 5 mL H_2_SO_4_ solution by stirring for one hour. Two identical slices of AgNWs@rGA were soaked in PVA-H_2_SO_4_ electrolyte for one hour and exhibited a large electric resistance of ≈188.6 Ω. Following, they were assembled into flexible symmetric AgNWs@rGA//AgNWs@rGA supercapacitor with a separator, placing at room temperature in the experimental fume hood for 36 h to let PVA-H_2_SO_4_ electrolyte natural solidification; the humidity was maintained at about 30% RH. Two polyester film (PET) slices wrapped outside were used as the package, and the electrode was led out with copper tape. Simultaneously, the charged symmetric AgNWs@rGA//AgNWs@rGA supercapacitor served as the sensitive piezoionic self-powering sensor.

### 2.3. Characterization

The structures and morphologies of AgNWs@rGA were obtained by X-ray powder diffractometer (XRD, SmartLab 9 KW, Rigaku, Tokyo, Japan) with wavelength of 1.541 Å, scanning electron microscope (SEM, Hitachi S-4800, Hitachi, Tokyo, Japan), and transmission electron microscopy (JEM 2100, JEOL Ltd., Tokyo, Japan). Raman spectra (Renishaw-inVia, Renishaw, Wotton-under-Edge, England) of AgNWs@rGA were carried out with a wavelength of 532 nm. The stress–strain performance of the AgNWs@rGA composite spring was obtained by an electronic material testing machine (UTM-2102, SUNS, Shenzhen, China). The current response of sensitive piezoionic self-powering sensor at pressure of 0–3.6 kPa was tested by connecting with a Keithley Model 2400 Source Meter with UTM-2102. Additionally, an electronic analytical balance (BSA124S-CW, Sartorius, Göttingen, Germany, Max = 120 g, d = 0.1 mg) was used for weighting.

## 3. Results and Discussion

The SEM of AgNWs@rGA shown in Figure 1a illustrates obvious cross-linking porous framework with flexibility and variability of pore size. Figure 1b confirms silver nanoparticles and nanowires are uniformly anchored on the surface of graphene without significant agglomeration, facilitating fast electronic transfer during pressure-sensing and electrochemical energy storage processes [15,16]. Figure 1c demonstrates that the XRD pattern of AgNWs@rGA comprises typical diffraction peak corresponding with (002) plane reflections of GA, and diffraction peaks at 38.1°, 44.3° plane reflections conforming to (111) and (200) plane of Ag [17]. The Raman spectra of AgNWs@rGA and GO is displayed in Figure 1d. All the samples present two vibrational bands closed to 1348 and 1583 cm^−1^, corresponding to D and G bands, respectively. The intensity ratio of D band to G band is an indicator of the degree of reduction and in-plane sp2 domains. The D/G intensity of AgNWs@rGA (1.04) is higher than that of GO (0.91), demonstrating an increase in defects after implanting with AgNWs and enhancing electron/ion transportation. The stress–strain performance of AgNWs@rGA is investigated under different strains of 10–90% in Figure 1e, demonstrating structural stability with almost complete recovery from 90% strain, and excellent elastic performance. Figure 1f presents the TEM of AgNWs@rGA, illustrating that AgNWs distribute evenly on the surface of graphene. HRTEM image in Figure 1g reveals clear lattice fringes of AgNWs with interspacing of 0.24 nm, corresponding to the (111) plane of metallic Ag. The energy dispersive spectroscopy mapping results further confirms the uniform distribution and existence of the Ag (Figure 1h) and C (Figure 1i) elements.

The electrochemical performance of compressive symmetric supercapacitor based on AgNWs@rGA electrode was investigated. Typical cyclic voltammetry (CV) curves of the device from 1.1 V to 1.5 V at 200 mV s^−1^ are shown in Figure 2a; the fabricated device presents quasi-rectangular characteristic under all the working voltage, indicating good capacitive behavior. As displayed in Figure 2a, no obvious changes in CV curves are observed below 1.4 V, but when the potential window increases to 1.5 V, the current increases greatly and the shape of CV curve changes, indicating the optimum potential of 1.5 V. Galvanostat charge-discharge (GCD) curves under different working voltage at 1.0 A g^−1^ in Figure 2b exhibit stable triangular shape, supporting the stable electrochemical voltage of AgNWs@rGA//AgNWs@rGA. Based on results of GCD, the specific capacitance of symmetric supercapacitor is illustrated in Figure 2c. The specific capacitance of AgNWs@rGA//AgNWs@rGA was calculated via the following equation:(1)$$C=\frac{I\times \Delta t}{m\times \Delta V}$$

Here, Δ*t* (s) represents discharging time, *I* (A) donates corresponding discharging current, *m* (g) is total mass of active material AgNWs@rGA, and Δ*V* (V) means corresponding voltage window. Specific capacitance gradually increases with the increase in potential windows. The device delivers the maximum specific capacitance of 146.6 F g^−1^ at 1.5 V. Figure 2d depicts CV curves of the symmetrical supercapacitor at a scan rate from 50 to 500 mV s^−1^, illustrating a quasi-rectangular shape even at high 500 mV s^−1^, disclosing a typical double-layer capacitance behavior and excellent rate capability [18]. The good symmetry of GCD curves from 1.0 to 5.5 A g^−1^ further confirm its efficient capacitive behavior in Figure 2e. The specific capacitances estimated from the GCD curves are summarized in Figure 2f. The maximum specific capacitance for symmetric supercapacitor reaches to 146.6 F g^−1^ at 1.0 A g^−1^, and then declines to 70.6 F g^−1^ at 5.5 A g^−1^. This decrease in specific capacitance stems from the insufficient sluggish ion diffusion under high current densities [19]. The outstanding performance stems from outstanding electron transport and ion transferring during electrochemical dynamics process. Cycling retention is significant electrochemical indicators as supercapacitor. As shown in Figure 2g, after 4000 cycles, a retention ratio as high as 80.2% is obtained, indicating an excellent cycling stability for AgNWs@rGA.

Utilizing the energy storage properties of supercapacitors, sensitive piezoionic self-powering sensor based on symmetric AgNWs@rGA//AgNWs@rGA structure was achieved. A schematic working illustration of the self-powered sensor is presented in Figure 3a. As a result, Figure 3b offers the relative change of current (Δ*I/I*_0_) under different applied pressures, where Δ*I* and *I*_0_ presents the change of current and initial current, respectively. Obviously, Δ*I/I*_0_ illustrates sensitive increasement with increasing the pressure. The repeatability test of Δ*I/I*_0_ under pressure of 0.8, 1.6, 2.4, and 3.2 Pa is shown in Figure 3c, exhibiting high reliability toward various pressure. Figure 3d presents the stable step increasement for Δ*I/I*_0_ responding to step-pressure, as well as stepped decreasing in the process of unloading without hysteresis. The pressure sensitivity (S) indicates the slope in curve of *ΔI/I_0_* with the applied pressure in Figure 3e. The sensor illustrates excellent linearity and sensitivity of 2.54 kPa^−1^ in the pressure range of 0–3.6 kPa, exhibiting better performance compared with that of previous reports as listed in Table 1. Such behavior can be explained in Figure 3f; the internal microstructural frame deforms under external pressure, a large pressure increases the electron transmission path and sufficient interface, decreasing the contact resistance and internal resistance. Moreover, pressure will force positive and negative electrolyte ions to redistribute at the interface between electrolyte and active material, creating a new electric double layer.

Figure 4a presents a schematic illustration of the assembled flexible symmetrically structure sensor. As shown in Figure 4b, the Δ*I/I*_0_ of the sensor remains almost independent with the applied frequency under the pressure of 1.6, 2.0, and 2.4 kPa, respectively. The device presents excellent stability and high durability during 1000 continued compressing–releasing cycles under the pressure of 2.4 kPa (Figure 4c). The SEM of AgNWs@rGA after stress–strain cycles is provided in the insets of Figure 4c, illustrating that the elastic composite aerogel generally presents an almost intact micromorphology, but some cracks and bending of the aerogel wall appear after multiple pressure cycles. These results confirm the highly sensitive and reproducible performance of the fabricated piezoionic self-powering sensor based on the symmetric AgNWs@rGA//AgNWs@rGA structure.

## 4. Conclusions

In conclusion, based on the highly compressible AgNWs@rGA//AgNWs@rGA, a novel piezoionic self-powered pressure sensor was designed. This ingenious construction integrates pressure sensing ability and energy storage function into a single flexible compressible device. Serving as energy device, it behaves with a prominent specific capacitance of 146.6 F g^−1^, and excellent rate capability even at 500 mV s^−1^. The fabricated piezoionic self-powered sensor exhibits a high sensitivity of 2.54 kPa^−1^, and superior pressure sensing stability up to 1000 compressive cycles. Our work provides a new research direction for the design integration of self-driving wearable electronics.

## Figures and Tables

**Figure 1 nanomaterials-12-02574-f001:**
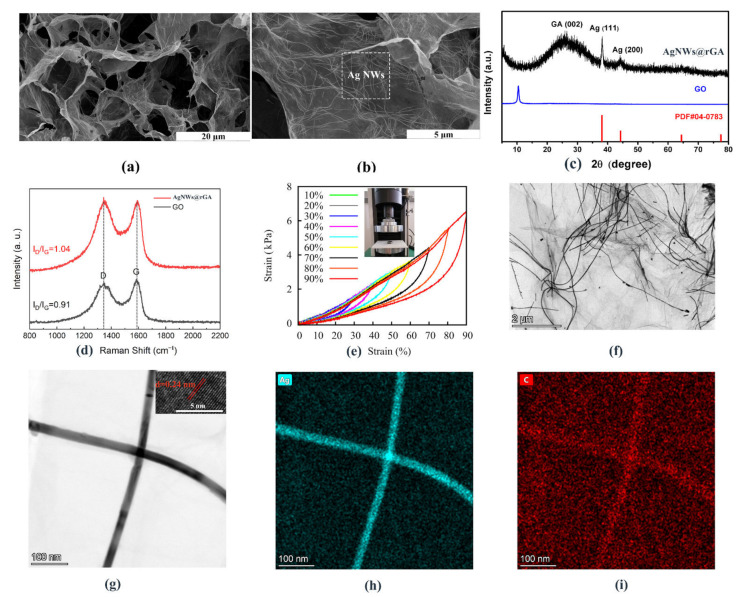
(**a**,**b**) SEM images of AgNWs@rGA composite aerogel spring. (**c**) XRD patterns of AgNWs@rGA together with pure GO. (**d**) Raman spectra of AgNWs@rGA. (**e**) Compressible stress–strain curves of AgNWs@rGA under strain of 10%, 20%, 30%, 40%, 50%, 60%, 70%, 80%, and 90%, respectively. The insets represent the photo image of AgNWs@rGA composite and testing device. (**f**) TEM images of AgNWs@rGA. (**g**) HRTEM image of AgNWs@rGA. (**h**,**i**) EDS mapping of the elements Ag and C.

**Figure 2 nanomaterials-12-02574-f002:**
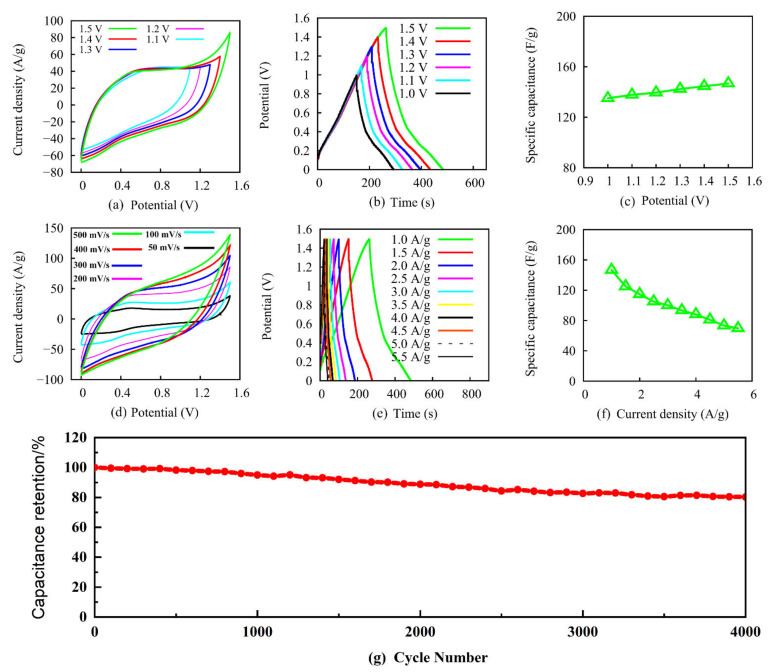
(**a**) CV curves of AgNWs@rGA/AgNWs@rGA supercapacitor under different voltage windows of 1.1, 1.2, 1.3, 1.4, and 1.5 V, respectively. (**b**) GCD curves of AgNWs@rGA//AgNWs@rGA under different voltage windows. (**c**) Specific capacitances versus scan voltage. (**d**) CV curves of AgNWs@rGA//AgNWs@rGA varying from 50 to 500 mV s^−1^. (**e**) GCD curves of AgNWs@rGA//AgNWs@rGA at different current densities of 1.0 A/g to 5.5 A/g. (**f**) Specific capacitances of AgNWs@rGA//AgNWs@rGA versus current density. (**g**) Cycling performance of AgNWs@rGA//AgNWs@rGA supercapacitor.

**Figure 3 nanomaterials-12-02574-f003:**
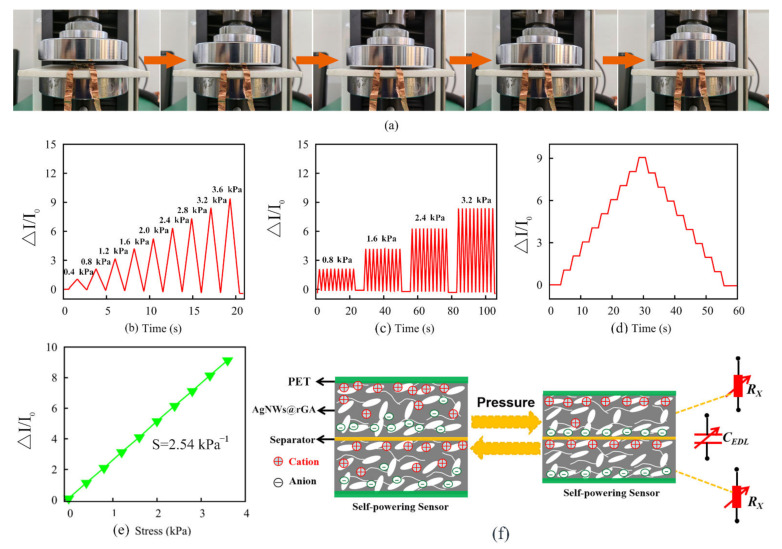
(**a**) Schematic working illustration of the self-powered sensor. (**b**) Δ*I/I*_0_ versus time plots under different pressure of 0–3.6 kPa. (**c**) Repeated measurements of Δ*I/I*_0_ of self-powering pressure sensor under different pressures of 0.8, 1.6, 2.4, and 3.2 kPa. (**d**) Δ*I/I*_0_ of the sensor toward stepped pressure of 0–3.6 kPa. (**e**) Δ*I/I*_0_ with the applied pressure, slope represents sensitivity S. (**f**) Working mechanism of piezoionic self-powering sensor.

**Figure 4 nanomaterials-12-02574-f004:**
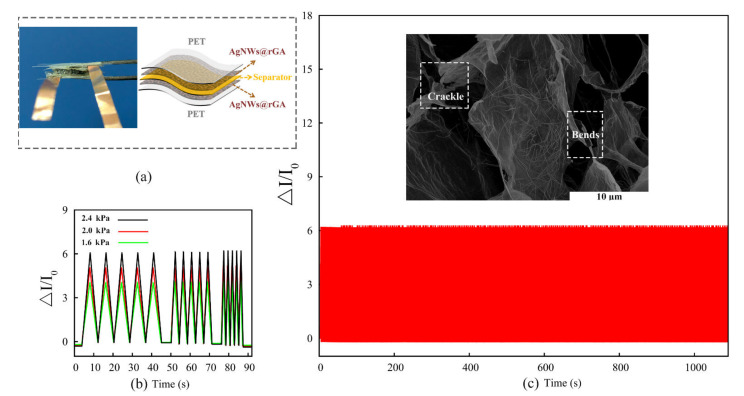
(**a**) Photograph of the assembled AgNWs@rGA//AgNWs@rGA wearable electronics and a graphic illustration of the piezoionic self-powering sensor. (**b**) Responding *ΔI/I_0_* under loading pressure of 1.6, 2.0, and 2.4 kPa with different frequencies. (**c**) Compressing–releasing cycles of piezoionic self-powering sensor under the pressure of 2.4 kPa. The inset is the SEM of AgNWs@rGA after 1000 stress–strain cycles.

**Table 1 nanomaterials-12-02574-t001:** Summary of the graphene-based pressure sensors compared with our work.

Material	Preparation Method	Sensitivity/GF	Detection Range	Stability	Refs
rGA	Solvothermal, freeze drying and high temperature annealing	0.46 kPa^−1^	0.5–8 kPa	4200	[20]
RGO-PU-HT	Dip coating and hydrothermal reduction	0.26 kPa^−1^	0201310 kPa	10,000	[21]
PDMS/EGO	Direct ink writing	0.31 kPa^−1^	0.248–500 kPa	1000	[22]
GO/Ag NWs	Dip coating	0.016 kPa^−1^	0–40 kPa	7000	[9]
MLG-PDMS	Drop casting	0.23 kPa^−1^	10–70 kPa	-	[23]
NGS	Hydrothermal and thermal annealing	1.33 kPa^−1^	30–40% (Strain)	3000	[24]
3D GF	Self-assembly	0.183 kPa^−1^	0–66 kPa	10,000	[25]
AgNWs@rGA	Self-assembly	2.54 kPa^−1^	0–3.6 kPa	1000	This work

## Data Availability

The data presented in this study are available on request from the corresponding author.

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
