# Peer review of "Highly Compressible Elastic Aerogel Spring-Based Piezoionic Self-Powering Pressure Sensor for Multifunctional Wearable Electronics"

_nanomaterials, 2022, doi:10.3390/nano12152574_

Round 1

Reviewer 1 Report

This work delivers a simple concept about a supercapacitor which can be used as a self-powered pressure sensor. While each concept of supercapacitor or pressure sensor based on nanocomposite material is not new, the multifunctional concept together with the fairly new material (Ag nanowire in graphene oxide aerogel) makes this work interesting for some readers. However, there must be some revisions before the publication.

1. The magnified device image after completing the fabrication should be provided.

2. For the specific capacitance measurement of Figure 2c, 2f, what value did you use for the normalization? (For example, total device weight, electrode weight, or active material (AgNPW@GA) weight). Please provide more details about the specific capacitance calculation.

3. Figure resolution and the font size must be improved. In particular, some inset fonts are not recognizable at all.

4. Grammar check throughout the manuscript is needed for better readability.

Author Response

Dear Editors and Reviewers:

Thank you very much for offering us an opportunity to revise our manuscript. We appreciate your works very much on our manuscript. We have revised the manuscript carefully following to your comments and suggestions. The revised parts are marked in red in the revision edition.

Reviewer #1: Comments and Suggestions for Authors

This work delivers a simple concept about a supercapacitor which can be used as a self-powered pressure sensor. While each concept of supercapacitor or pressure sensor based on nanocomposite material is not new, the multifunctional concept together with the fairly new material (Ag nanowire in graphene oxide aerogel) makes this work interesting for some readers. However, there must be some revisions before the publication.

  1. The magnified device image after completing the fabrication should be provided.

Answer: We have provided the magnified device image of self-powered pressure sensor in Fig. 4(a).

  1. For the specific capacitance measurement of Figure 2c, 2f, what value did you use for the normalization? (For example, total device weight, electrode weight, or active material (AgNPW@GA) weight). Please provide more details about the specific capacitance calculation.

Answer: The calculation of mass specific capacitance in Figure 2c, 2f is based on the mass of active material AgNWs@rGA. According to our mature and stable prepared process, the average quality of aerogel gel is controlled at about 15.4 mg. Then, two slices of aerogel with equal mass of about 1.8 mg are cut off from the monoblock of AgNWs@rGA. Then, the all-solid-state supercapacitor is assembled by placing two slices of AgNWs@rGA with PVA-H2SO4 as gel electrolyte, and two polyester film (PET) slices wrapped outside as the package. Besides, we have provided the detailed specific capacitance calculation for the supercapacitor in page 7 line 6-11 and marked in red.

  1. Figure resolution and the font size must be improved. In particular, some inset fonts are not recognizable at all.

Answer: Thank you for your suggestions, we have checked and revised the Figure resolution and the font size for all the Figures to be recognizable. Besides, according to the quality requirements of Nanomaterials for submitted pictures, the pixels of all submitted pictures have been set to 600 dpi.

  1. Grammar check throughout the manuscript is needed for better readability.

Answer: We have revised grammar errors throughout the manuscript carefully and marked in red.

Reviewer #2:

This paper deals with the development of a self-powering multifunctional pressure sensors and supercapacitors integrated device based on highly elastic silver nanowires@graphene aerogel spring. This work contains interesting results but their presentation is rather succinct. Moreover, the discussion of the results is almost non-existent. The authors did not clearly provide the objective of their work. I consider that a major revision is required before publication.

  1. Overall, the English is not satisfactory. Please revise.

Answer: Thank you for your suggestions, we have carefully revised grammar errors and improved the English with the help of experts with good English in the manuscript which are marked in red.

  1. Figure captions are not satisfactory. Figures are stand-alone sections. Therefore, they should provide all the required information in the captions such that the reader is not required to refer to the main text. Figure captions should be included below each figure.

Answer: Thank you for your suggestions, we have revised Figure captions and rearranged the Figures as stand-alone sections which are marked in page 11-14 in red.

  1. Please define acronym when appeared for the first time in the text.

Answer: Thank you for your suggestions, we have rechecked all the acronym and provided corresponding full name for the first time, which are marked in paper.

  1. Page 2, top: if KPa means kilo Pascal, it should be written with small k as kPa.

Answer: Thank you for your suggestions, we have revised the unit KPa to be kPa in paper and all the Figures.

  1. Section 2.3: please provide X-ay wavelength

Answer: Thank you for your suggestions, we have provided the X-ay wavelength of 1.541 Å in Section 2.3 and marked in page 5 line 10 in red.

  1. Page 2, line 27: it should be “SEM of AgNPW@GA shown in Fig…”

Answer: Thank you for your suggestions, we have revised this sentence to be “SEM of AgNWs@rGA shown in Fig. 1(a) illustrates obvious cross-linking porous framework with flexibility and variability of pore size”, which are marked in page 5 line 18-19 in red.

  1. The graphene aerogel is not well characterized. Please provide Raman spectrum showing the D- and G-bands of carbon and discuss their relative intensity accordingly.

Answer: Thank you for your suggestions, we have provided Raman spectrum to discuss the D- and G-bands of carbon in Fig. 1(d). Moreover, we have added the corresponding discussions in page 6 in red in details.  

  1. Fig. 1c: the standard diffractogram of GO is useless because no reflection is observed.

Answer: Thank you for your suggestions, the diffraction peak of GO is not observed in AgNWs@rGA composite aerogel, illustrating that GO nanosheets is reduced to rGO during self-assembly. GO displays a sharp and intense diffraction peak at 2θ= which ascribes to the (001) lattice plane, corresponding to a d-spacing value of 0.85 nm between the stacked GO sheets. After the reduction of GO, this sharp peak disappears and AgNWs@rGA shows a strong diffraction peak at 2θ=, which is attributed to the (002) plane with an interlayer d-spacing of 0.33 nm which attributes to the well interaction of AgNWs with the rGO sheet. Also, the peak broadening indicates the successful prevention of the rGO restacking due to the insertion of AgNWs, which is in well agreement with the literature reports.

  1. For clarity, Figures 2 should be enlarged

Answer: Thank you for your suggestions, Figures 2 have been enlarged and the pixels have been set to 600 dpi.

  1. A discussion is missing for the comparison of the present sensors with similar devices from literature. Please provide a comparison chart.

Answer: Thank you for your suggestions, we have provided the comparison of the present sensors with similar devices from literature and marked in page 8 line 16-17 in red. Besides, we have provided the corresponding comparison chart, which are marked in page 15 in red.

  1. References: please provide title of articles.

Answer: Thank you for your suggestions, we have provided the title of articles for references.

Reviewer #3: The authors have presented a single active material (AgNPW@GA) based multifunctional device (pressure sensor & supercapacitor), the prospect of which seems promising, and it has potential for practical applications. However, there are significant gaps in research design and execution and presentation of the results. The reviewer points out the following questions but suggests that the article be rejected.

  1. The abbreviation, AgNPW@GA of the main active material is never properly introduced. What does this stand for?

Answer: Thank you for your suggestions, we have revised the abbreviation AgNPW@GA to be AgNWs@rGA in order to express the material composition more clearly. AgNWs@rGA stands for 3D silver nanowire/reduced graphene aerogel, silver nanowires can maintain their original good morphology during self-assembly, while GO nanosheets is reduced to rGO during self-assembly. The final silver nanowire @reduced graphene composite aerogel is abbreviated as AgNWs@rGA.

  1. The authors mentioned a symmetric supercapacitor with two monoblocks of active material aerogel spring immersed and dried in gel electrolyte. But they never explained how they assembled the supercapacitor.

Answer: We have provided the detailed process of assembling the symmetric supercapacitor in Section 2.2 Preparation of Flexible Symmetrical Supercapacitor and Sensitive Piezoionic Self-powering Sensor in red. The symmetrical supercapacitor is fabricated as follows. The all-solid-state supercapacitor is assembled by placing two slices of aerogel with equal mass of about 1.8 mg which are cut off from the monoblock of AgNWs@rGA. PVA-H2SO4 is used as gel electrolyte, which is prepared as follows: 6 g of PVA is dissolved in 60 mL of water at 90 °C under stirring until solution becomes transparent. Then, dissolved PVA is added with 5 ml H2SO4 solution (3 M) by stirring for one hour. Two identical monoblocks of AgNWs@rGA are soaked in PVA-H2SO4 gel electrolyte for one hour. Following, they are assembled into flexible symmetric AgNWs@rGA//AgNWs@rGA supercapacitor with a separator, placing at room temperature in the experimental fume hood for 36 h to let PVA-H2SO4 electrolyte natural solidification, the humidity is maintained at about 30 %RH. Two polyester film (PET) slices wrapped outside is used as the package, the electrode is led out with copper tape. Simultaneously, the charged symmetric AgNWs@rGA//AgNWs@rGA supercapacitor is serving as the sensitive piezoionic self-powering sensor. 

  1. Apart from XRD and SEM, there is no physical characterization done to confirm the formation of the Ag nanowire-based aerogel.

Answer: Thank you for your suggestions, we have provided the TEM (Fig. 1(f)), HRTEM (Fig. 1(g)) as well as element mapping (Fig. 1(h)-(i)) for AgNWs@rGA to further confirm the formation of the Ag nanowire-based aerogel. Moreover, we have added the discussions in page 6 line 12-17 in red in details.  

  1. A stable operating window of an aqueous supercapacitor can never extend beyond 1.23 V without the possibility of electrolysis (oxygen evolution reaction). Therefore, the figure 2c showing improvement in specific capacitance with increasing potential, is only coming from the increased area under the curve due to some faradaic contribution, like OER, that can’t be considered as capacitive contribution. There is no justification on why they have chosen the optimum potential window as 1.5V.

Answer: The prepared flexible symmetric AgNWs@rGA//AgNWs@rGA supercapacitor adopts the all-solid-state PVA-H2SO4 as electrolyte, therefore, the stable working voltage window of those symmetrical supercapacitors is allowed to exceed 1.23 V, such as Advanced Energy Materials, 2021, 11(26): 2100768, Advanced Functional Materials, 2018, 28(44): 1805618, Chemical Engineering Journal, 2019, 371: 679-692, Chemical Engineering Journal 360 (2019): 171-179, ACS Applied Energy Materials, 2020, 3(9): 9379-9389. We choose the optimum potential window as 1.5 V for our supercapacitor attributed to the following factors. As shown in Fig. 2(a), the CV curves of AgNWs@rGA//AgNWs@rGA supercapacitor with various working voltage windows at 200 mV s−1 are collected. As displayed in Fig. 2(a), no obvious changes in CV curves are observed even when the potential window increased to 1.4 V, but when the potential window increases to 1.5 V, the current increases greatly and the shape of the CV curve changes, indicating that the supercapacitor can only be stable under the potential window no more than 1.5 V. Additionally, Fig. 2(d) illustrates that the shape of CV curves at the potential window of 1.5 V under the scan rates of 50–500 mV s−1 can be well maintained, indicating the device has good capacitive performance. In addition, the GCD curves at the potential window of 1.5 V of supercapacitor show symmetric shapes with slight deviations from linearity at various current densities from 1 to 5.5 A g−1 in Fig. 2(e). Therefore, 1.5 V is selected as the potential window and electrochemical performance are tested. This has also been adopted widely by other previous reports, such as ACS Appl. Energy Mater. 2020, 3, 9379−9389, Chemical Engineering Journal 409 (2021) 127891. Moreover, we have added the discussions in page 7 line 1-3 in red.  

  1. AgNW based graphene aerogels are not novel as materials, they have been reported before even as stretchable supercapacitors, but with better electrochemical performance.

Answer: Thank you for your suggestions, AgNWs-based graphene aerogels have been reported. While in our work we introduce a novel multifunctional pressure sensors and all-solid-state flexible supercapacitors integrated device based on highly elastic silver nanowires@graphene aerogel spring. Existing reports achieve the multifunction of electronic devices, while the energy-storing and sensing functions are realized independently with different device structures. The implementation of various device structures increases the integration difficulty and device size. The designing and fabrication of highly elastic silver nanowires@graphene aerogel spring-based device in our work can realize both pressure sensing and energy storage functions into single structure.

  1. Although through the introduction, the authors claim to present a single structure based multifunctional device, they have only shown individual performance of a supercapacitor and a pressure sensor of the same active material. There is no practical demonstration of how these two separate energy storage and sensor system would perform in an integrated device.

Answer: Thank you for your suggestions, here we lie in designing and fabrication of highly elastic silver nanowires@graphene aerogel spring-based device that can realize multifunctional pressure sensing and energy storage functions independently. The working diagram is shown as below. Moreover, we have provided the working photograph for AgNWs@rGA//AgNWs@rGA self-powering sensor in Fig. 3(a).

  1. Stability of both storage and sensor system is an important parameter to consider while assembling such devices, the authors did not show any cycling stability performance of the supercapacitor system. Stability of the sensor system needs to be demonstrated beyond stress-strain cycles, i.e., through presentation of physical characterizations. It needs to be shown how after continuous cycling in both storage and sensing systems, if there is any significant morphological changes in the active material.

Answer: We have provided the cycling stability performance of the supercapacitor system in Fig. 2(g), and added the discussions in page 8 line 1-3 in red. Moreover, we have provided the SEM of AgNWs@rGA after 1000 stress-strain cycles to illustrate any significant morphological changes among the active material in the insets of Fig. 4(c) and added the discussions in page 9 line 6-10 in red.

Reviewer #4: Prior to acceptance the following changes are required.

  1. Introduction:

The introduction should provide with more background on energy devices or pressure sensing based on similar materials in order to assess the performance as single device. Please provide with performance of similar integrated pressure sensor and energy devices.

Answer: Thank you for your suggestions, we have provided the performance of similar integrated pressure sensor and energy devices in introduction part in page 2 in red.

  1. Experimental:

2.1 What is silver nanowires solution?? What is composition, solvent etc??

Answer: Thank you for your suggestions, silver nanowires solution (5 mg mL-1) is composed of silver nanowires with a length of 18-23 µm and diameter in the range of 25-30 nm, which are dispersed in solvent water. Moreover, we have added those in Section 2.1 Synthesis of Silver Nanowires@Graphene Aerogel in red.

2.2 Please provide with autoclave dimensions.

Answer: Thank you for your suggestions, we have provided with autoclave dimensions (25 mL) in Section 2.1 Synthesis of Silver Nanowires@Graphene Aerogel in red.

2.3 Please provide with dimensions of initial dispersion, hydrogel and aerogel (monoblock).

Answer: Thank you for your suggestions, we have provided with dimensions of initial dispersion (4 mL), hydrogel with an average height of about 5.2 mm and diameter of about 14.5 mm, and aerogel (monoblock) with an average thickness of about 4.6 mm and diameter of about 13.8 mm in Section 2.1 Synthesis of Silver Nanowires@Graphene Aerogel in red.

2.4 Please provide freeze-drying conditions and rate.

Answer: Thank you for your suggestions, we have provided the freeze-drying conditions in Section 2.1 Synthesis of Silver Nanowires@Graphene Aerogel in red.

2.5 Section 2.2: what it means ‘placing for 36h to let .. solidification’ - placing where? What conditions?

Answer: Thank you for your suggestions, the device is placed at room temperature in the experimental fume hood for 36 h to let PVA-H2SO4 electrolyte natural solidification. Besides, we have provided the conditions in page 5 line 1-4 in red.

2.6 Please provide with elements of the device fig 3e indicates a PTE element which is not described)

Answer: Thank you for your suggestions, we make a mistake here, two PET slices wrapped outside is used as the package instead of PTE, we have revised PTE to be PET. Besides, PET film also known as polyester film, can achieve high conductivity by depositing ITO on its surface.

  1. Results:

3.1 Authors indicate in fig 1b the synthesis of both Ag nanoparticles and nanowires, however the abstract is missleading as it mentions ‘nanowire’, while in the experimental they abbreviate the material with AgNWs@rGA that shows both forms (by the way the abbreviation of material is not described) while in figures 3 and 4 they mention a different abbreviation- AWGA (not described previously either).

Answer: Thank you for your suggestions, we have revised the abbreviation AgNPW@GA to be AgNWs@rGA in order to express the material composition more clearly. AgNWs@rGA stands for 3D silver nanowire/reduced graphene aerogel, silver nanowires can maintain their original good morphology during self-assembly, while GO nanosheets is reduced to rGO during self-assembly. The final silver nanowire @reduced graphene composite aerogel is abbreviated as AgNWs@rGA. Besides, we have removed the different abbreviation-AWGA in Fig. 4(a).

3.2 How do the authors distinguish from the effect of both morphologies on either device performance?? The authors should provide with elemental mapping to assess the distribution of Ag nanostructures.

Answer: Thank you for your suggestions, we have provided the elemental mapping to assess the distribution of Ag nanostructures in Fig. 1(h) and Fig. 1(i), and the TEM (Fig. 1(f)) as well as SEM (the insets in Fig. 4(c)) of AgNWs@rGA after 1000 stress-strain cycles to illustrate any significant morphological changes in the active material. Moreover, we have added the discussions in page 6 line 12-17 in red in details.

3.3 Please address English (syntax, typos etc).

Answer: Thank you for your suggestions, we have revised grammar errors in the manuscript carefully and marked in red.

Reviewer 2 Report

see attached file

Author Response

(The authors gave the same response as above.)

Reviewer 3 Report

The authors have presented a single active material (AgNPW@GA) based multifunctional device (pressure sensor & supercapacitor), the prospect of which seems promising, and it has potential for practical applications. However, there are significant gaps in research design and execution and presentation of the results. The reviewer points out the following questions but suggests that the article be rejected.

(1)   The abbreviation, AgNPW@GA of the main active material is never properly introduced. What does this stand for?

(2)   The authors mentioned a symmetric supercapacitor with two monoblocks of active material aerogel spring immersed and dried in gel electrolyte. But they never explained how they assembled the supercapacitor.

(3)   Apart from XRD and SEM, there is no physical characterization done to confirm the formation of the Ag nanowire-based aerogel.

(4)   A stable operating window of an aqueous supercapacitor can never extend beyond 1.23 V without the possibility of electrolysis (oxygen evolution reaction). Therefore, the figure 2c showing improvement in specific capacitance with increasing potential, is only coming from the increased area under the curve due to some faradaic contribution, like OER, that can’t be considered as capacitive contribution. There is no justification on why they have chosen the optimum potential window as 1.5V.

(5)   Ag NW based graphene aerogels are not novel as materials, they have been reported before even as stretchable supercapacitors, but with better electrochemical performance.

(6)   Although through the introduction, the authors claim to present a single structure based multifunctional device, they have only shown individual performance of a supercapacitor and a pressure sensor of the same active material. There is no practical demonstration of how these two separate energy storage and sensor system would perform in an integrated device.

()     Stability of both storage and sensor system is an important parameter to consider while assembling such devices, the authors did not show any cycling stability performance of the supercapacitor system. Stability of the sensor system needs to be demonstrated beyond stress-strain cycles, i.e., through presentation of physical characterizations. It needs to be shown how after continuous cycling in both storage and sensing systems, if there is any significant morphological changes in the active material.

Author Response

(The authors gave the same response as above.)

Reviewer 4 Report

Prior to acceptance the following changes are required.

Introduction: 

The introduction should provide with more background on energy devices or pressure sensing based on similar materials in order to assess the performance as single device. Please  provide with performance of similar integrated pressure sensor and energy devices.

Experimental:

What is silver nanowires solution?? What is composition, solvent etc??

Please provide with autoclave dimensions.

Please provide with dimensions of initial dispersion, hydrogel and aerogel (monoblock).

Please provide freeze-drying conditions and rate.

Section 2.2: what it means ‘placing for 36h to let .. solidification’ - placing where? What conditions?

Please provide with elements of the device fig 3e indicates a PTE element which is not described)

Results: 

Authors indicate in fig 1b the synthesis of both Ag nanoparticles and nanowires, however the abstract is missleading as it mentions ‘nanowire’, while in the experimental they abbreviate the material with AgNPW@GA that shows both forms (by the way the abbreviation of material is not described) while in figures 3 and 4 they mention a different abbreviation- AWGA (not described previously either).

How do the authors distinguish from the effect of both morphologies on either device performance?? The authors should provide with elemental mapping to assess the distribution of Ag nanostructures. 

Please address English (syntax, typos etc).

Author Response

(The authors gave the same response as above.)

Round 2

Reviewer 2 Report

The authors have addressed my comments satisfactorily. I recommend publication of this work.

Author Response

Thanks a lot for your efforts, and appreciate your works very much on our manuscript.

Reviewer 3 Report

The authors have presented a thorough revision of their previous manuscript and the current version seems much improved. The reviewer still has some confusions regarding some discussions provided by the authors and would be very pleased if those are answered, it would eventually benefit the outcome of the manuscript. The current manuscript can be accepted after those minor corrections suggested by the reviewer.

(1)   In the solid-state fabrication section, it is mentioned that two identical slices of aerogels weighing about 1.8 mg were used as individual electrodes. Was a normal electronic weighing scale was used for this measurement? Because it increases the scale of error in weight measurement significantly as aerogels are extremely light weight materials. In that case, the authors are suggested to repeat the measurement at least for a significant number of samples and provide a calibration plot with error bar. Or at least provide the accuracy limit of the weighing instrument.

(2)   It would be interesting to see how the conductivity of the aerogels are impacted after dipping them in PVA gel for such a long time (36 h). The authors are suggested to do the pre and post fabrication conductivity study of the electrodes and provide the data.

(3)   The authors have mentioned in one of the comments, “… PET film also known as polyester film, can achieve high conductivity by depositing ITO on its surface…”. Is this how they have assembled the present all solid-state SC? If so, it is not mentioned the fabrication details. Please clarify.

(4)   The reviewer wants to thank the authors for introducing some very interesting articles that have in fact reported aqueous solid-state supercapacitors beyond the oxygen evolution potential limit. It is in fact argued (Advanced Functional Materials, 2018, 28(44): 1805618) that the achievement of 1.4 V potential window was possible because “when the full cell was charged to 1.4 V, the cell reached the upper limit of 1.2 V; ascribing such high voltage operation to enlarged maximum cut-off voltage of the electrode”; which seems like a logical explanation.

Here the reviewer wants to point out that the authors in Advanced Energy Materials, 2021, 11(26): 2100768 have avoided the use of aqueous electrolyte and opted for LiCl to avoid the issue of water splitting and dissolution of electrode materials in electrolyte leading to poor stability. Almost all the other cited articles (ACS Applied Energy Materials, 2020, 3(9): 9379-9389; Advanced Functional Materials, 2018, 28(44): 1805618) have mentioned that at the potential where the anode current increases significantly and shape of the CV curve changes, it clearly indicates the onset of oxygen evolution and therefore, the device should be operated till the potential until there is no significant changes in the CV curve. Similarly, in your case, the significant increase in anodic current is clearly visible at 1.5 V, meaning you cannot broaden the window beyond 1.4 V without the risk of OER. The argument that the author provided therefore is misleading. Please explain or correct the potential limit up to 1.4 V and calculate the corresponding storage parameters.

(5) Practical demonstration of a self-powered active pressure sensor is still missing. The working photograph for AgNWs@rGA//AgNWs@rGA self-powering sensor provided in Fig. 3(a) does not demonstrate any storage activity. It would be nice to provide a real time video on how the material is self-charging while working as a pressure sensor.

Author Response

Dear Editors and Reviewers:

Thank you very much for offering us an opportunity to revise our manuscript again. We appreciate your works very much on our manuscript. We have revised the manuscript carefully following to your comments and suggestions. The revised parts are marked in red in the revision edition.

Reviewer #3: The authors have presented a thorough revision of their previous manuscript and the current version seems much improved. The reviewer still has some confusions regarding some discussions provided by the authors and would be very pleased if those are answered, it would eventually benefit the outcome of the manuscript. The current manuscript can be accepted after those minor corrections suggested by the reviewer. 

(1)   In the solid-state fabrication section, it is mentioned that two identical slices of aerogels weighing about 1.8 mg were used as individual electrodes. Was a normal electronic weighing scale was used for this measurement? Because it increases the scale of error in weight measurement significantly as aerogels are extremely light weight materials. In that case, the authors are suggested to repeat the measurement at least for a significant number of samples and provide a calibration plot with error bar. Or at least provide the accuracy limit of the weighing instrument.

Answer: Thank you for your suggestions, an electronic analytical balance (BSA124S-CW, Sartorius, Max=120 g, d=0.1mg) is used for this measurement, we have provided the accuracy limit of the weighing instrument in page 5 and marked in red. Besides, two identical slices of aerogels weighing about 1.8 mg are cutting from the monoblock of AgNWs@rGA, we have provided a calibration plot with error bar for weighing aerogels below.

(2)   It would be interesting to see how the conductivity of the aerogels are impacted after dipping them in PVA gel for such a long time (36 h). The authors are suggested to do the pre and post fabrication conductivity study of the electrodes and provide the data.

Answer: Thank you for your suggestions, we have provided the pre and post fabrication conductivity study of the electrodes. The initial AgNWs@rGA aerogels exhibits good electric resistance of ≈22.5 Ω. After post fabrication, the soaked AgNWs@rGA aerogels exhibits large electric resistance of ≈188.6 Ω due to covered with a layer of PVA-H2SO4 gel electrolyte with poor conductivity.

(3)   The authors have mentioned in one of the comments, “… PET film also known as polyester film, can achieve high conductivity by depositing ITO on its surface…”. Is this how they have assembled the present all solid-state SC? If so, it is not mentioned the fabrication details. Please clarify.

Answer: The all-solid-state supercapacitor is fabricated using poly(vinyl alcohol) (PVA) phosphoric acid (H2SO4) gel electrolyte sandwiched between two identical slices of AgNWs@rGA aerogels electrodes, and two conductive ITO-PET films as the package. PVA-H2SO4 is used as gel electrolyte, which is prepared as follows: 6 g of PVA is dissolved in 60 mL of water at 90 °C under stirring until solution becomes transparent. Then, dissolved PVA is added with 5 ml H2SO4 solution by stirring for one hour. After being fully mixed, two identical slices of AgNWs@rGA are soaked in PVA-H2SO4 electrolyte to let the electrolyte fully immerse. Following, the two identical slices of AgNWs@rGA are painted on the ITO-PET film with silver paste and attached to the copper wires to fabricate a symmetrical all-solid-state supercapacitor device, placing at room temperature in the experimental fume hood for 36 h to let PVA-H2SO4 electrolyte natural solidification. We have provided the the fabrication details in page 4-5 in red in details.

(4)   The reviewer wants to thank the authors for introducing some very interesting articles that have in fact reported aqueous solid-state supercapacitors beyond the oxygen evolution potential limit. It is in fact argued (Advanced Functional Materials, 2018, 28(44): 1805618) that the achievement of 1.4 V potential window was possible because “when the full cell was charged to 1.4 V, the cell reached the upper limit of 1.2 V; ascribing such high voltage operation to enlarged maximum cut-off voltage of the electrode”; which seems like a logical explanation.

Answer: Thank you for your suggestions. Enlarging the electrochemical stable potential window of electrolyte can effectively improve the electrochemical performance of supercapacitors. While, the potential window of aqueous electrolyte is essentially limited to ≈0–1.2 V due to the theoretical decomposition voltage of water (1.23 V), referring to Advanced Functional Materials (2022): 2201166. The symmetric two-electrode design can enlarge the working voltage to a certain extent for aqueous supercapacitors (≈0–1.75 V), referring to ACS Nano 2020, 14, 7308; Small 2020, 16, 2004188; Adv. Energy Mater. 2018, 8, 1800408. In this work, the prepared all-solid-state supercapacitor is assembled with AgNWs@rGA slice as both positive and negative electrode, all the electrochemical measurements are carried out in a two-electrode system instead of a three-electrode system. Besides, the electrolytes are adopted all all-solid-state electrolytes instead of aqueous electrolytes, being conductive to reduce the risk of OER. There are many reports about the normal working voltage window of all solid-state symmetrical supercapacitors exceeding 1.4 V, such as Chemical Engineering Journal 436 (2022): 135231; ACS Applied Materials & Interfaces 13.44 (2021): 52519-52529; Advanced Functional Materials, 2018, 28(52): 1806207.

(5) Practical demonstration of a self-powered active pressure sensor is still missing. The working photograph for AgNWs@rGA//AgNWs@rGA self-powering sensor provided in Fig. 3(a) does not demonstrate any storage activity. It would be nice to provide a real time video on how the material is self-charging while working as a pressure sensor.

Answer: There is a misunderstanding here, we lie in designing and fabrication of highly elastic silver nanowires@graphene aerogel spring-based device that can realize dual-mode pressure sensing and energy storage functions, not meaning that pressure sensing and energy storage function work at the same time. The sensitive self-powering pressure sensor is realized based on self-discharging feather of flexible all-solid-state symmetrical supercapacitor, working independently with energy storage functions. We have provided a real time video on the working of sensor.

Reviewer 4 Report

The authors have improved the manuscript considerably. However, for one concern, they chose to simply delete the info. How can they distinguish the effect of nanowires and nanoparticles?

Author Response

Dear Editors and Reviewers:

Thank you very much for offering us an opportunity to revise our manuscript again. We appreciate your works very much on our manuscript. We have revised the manuscript carefully following to your comments and suggestions. The revised parts are marked in red in the revision edition.

Reviewer #4:

The authors have improved the manuscript considerably. However, for one concern, they chose to simply delete the info. How can they distinguish the effect of nanowires and nanoparticles?

Answer: Thank you for your suggestions. Generally, graphene aerogel exhibits poor electrical conductivity, being not conducive to electronic transmission. Besides, the efficient electron and ion transport pathways are related to the network structure. However, the prone to re-stacking of fully delaminated graphene flakes subjected to the great Van Edward force between them usually leads to inadequate exposure of active sites for charge storage, resulting in underdeveloped areal capacitance with tight stacking structure. The insertion of silver line not only can serve as anchors to strengthen the connection strength between 2D few-layered graphene flakes, resulting in enhanced mechanical behavior of graphene/Ag nanowire composite aerogel, but also serve as interlayer scaffolding to construct the ions channel between 2D few-layered graphene sheets, simultaneously alleviating the re-stack of 2D few-layered graphene sheets and guaranteeing more interlayer free voids for the easier transport of ions. The conductive silver nanowire anchored on the surface of graphene can serve as an effective electronic transmission path between propped-opened graphene flakes, in favor of increasing more active sites and charge storage. The synergistic action of silver nanowire and graphene achieves the design and construction for ions/electrons double transport channels in hierarchical porous aerogel framework. This feather facilitates electrolyte penetration and ensures electron transfer between layers, effectively increasing ion diffusion. Hence, insertion of silver nanowire concurrently realizes the regulation of interlayer conductivity and space for favorable electrons transport and ion intercalation between interlayers, which further promotes charge storage with multiple electrolyte channels. Besides, a large number of silver nanoparticles anchored on the surface of graphene are conductive to provide sufficient active sites in the electrochemical process, which has been illustrated by reference Adv. Funct. Mater. 2017, 1700041.
